# A Single Shot Pre-fusion-Stabilized Bovine RSV F Vaccine is Safe and Effective in Newborn Calves with Maternally Derived Antibodies

**DOI:** 10.3390/vaccines8020231

**Published:** 2020-05-18

**Authors:** Sabine Riffault, Sara Hägglund, Efrain Guzman, Katarina Näslund, Luc Jouneau, Catherine Dubuquoy, Vincent Pietralunga, Daphné Laubreton, Olivier Boulesteix, David Gauthier, Aude Remot, Abdelhak Boukaridi, Alexander Falk, Ganna Shevchenko, Sara Bergström Lind, Karin Vargmar, Baoshan Zhang, Peter D. Kwong, María Jose Rodriguez, Marga Garcia Duran, Isabelle Schwartz-Cornil, Jean-François Eléouët, Geraldine Taylor, Jean François Valarcher

**Affiliations:** 1University Paris-Saclay, INRAE, UVSQ, VIM, 78350 Jouy-en-Josas, France; luc.jouneau@inrae.fr (L.J.); catherine.dubuquoy@gmail.com (C.D.); Vincent.Pietralunga@gmail.com (V.P.); daphne.laubreton@inserm.fr (D.L.); isabelle.schwartz@inrae.fr (I.S.-C.); jean-francois.eleouet@inrae.fr (J.-F.E.); 2Host Pathogen Interaction Group, Unit of ruminant medicine, Department of Clinical Sciences, Swedish University of Agricultural Sciences, Box 7054, 75007 Uppsala, Sweden; sara.hagglund@slu.se (S.H.); katarina.naslund@slu.se (K.N.); jean-francois.valarcher@slu.se (J.F.V.); 3The Pirbright Institute, Ash Road, Pirbright, Woking, Surrey GU24 0NF, UK; e.guzman@oxb.com (E.G.); geraldine.taylor@pirbright.ac.uk (G.T.); 4INRAE, PFIE, 37380 Nouzilly, France; Olivier.boulesteix@inrae.fr (O.B.); david.gauthier@inrae.fr (D.G.); 5INRAE, University of Tours, ISP, 37380 Nouzilly, France; aude.remot@inrae.fr; 6University Paris Saclay, INRAE, AgroParisTech, GABI, 78350 Jouy-en-Josas, France; abdelhak.boukadiri@inrae.fr; 7Department of Chemistry-BMC, Uppsala University, 875007 Uppsala, Sweden; alexander.falk@kemi.uu.se (A.F.); ganna.shevchenko@kemi.uu.se (G.S.); sara.lind@kemi.uu.se (S.B.L.); 8Department of Biomedicine and veterinary public Health, Swedish University of Agricultural Sciences, Box 7054, SE-756 51, 875007 Uppsala, Sweden; karin.vargmar@slu.se; 9Vaccine Research Center, National Institute of Allergy and Infectious Diseases, National Institutes of Health, Bethesda, MD 20892, USA; baoshan.zhang@nih.gov (B.Z.); pdkwong@nih.gov (P.D.K.); 10Applied Immunology and Genetics, S.L. (INGENASA), 28037 Madrid, Spain; Mjrodriguez@ingenasa.com (M.J.R.); mgarcia@ingenasa.com (M.G.D.)

**Keywords:** bovine, neonate, pre-fusion conformation, respiratory syncytial virus, subunit vaccine

## Abstract

Achieving safe and protective vaccination against respiratory syncytial virus (RSV) in infants and in calves has proven a challenging task. The design of recombinant antigens with a conformation close to their native form in virus particles is a major breakthrough. We compared two subunit vaccines, the bovine RSV (BRSV) pre-fusion F (preF) alone or with nanorings formed by the RSV nucleoprotein (preF+N). PreF and N proteins are potent antigenic targets for neutralizing antibodies and T cell responses, respectively. To tackle the challenges of neonatal immunization, three groups of six one-month-old calves with maternally derived serum antibodies (MDA) to BRSV received a single intramuscular injection of PreF, preF+N with Montanide^TM^ ISA61 VG (ISA61) as adjuvant or only ISA61 (control). One month later, all calves were challenged with BRSV and monitored for virus replication in the upper respiratory tract and for clinical signs of disease over one week, and then post-mortem examinations of their lungs were performed. Both preF and preF+N vaccines afforded safe, clinical, and virological protection against BRSV, with little difference between the two subunit vaccines. Analysis of immune parameters pointed to neutralizing antibodies and antibodies to preF as being significant correlates of protection. Thus, a single shot vaccination with preF appears sufficient to reduce the burden of BRSV disease in calves with MDA.

## 1. Introduction

The orthopneumovirus respiratory syncytial virus (RSV) causes severe respiratory infections, mostly affecting infants and the elderly [1,2]. RSV is widespread around the world, and virtually every child has been infected with this virus by the age of two [3,4]. About 1/3 of RSV infections spread to the lower respiratory tract, causing bronchiolitis and pneumonia. As a result of RSV infections, between 1 and 3% of children in the developed world are hospitalized in pediatric intensive care units in the winter season, and >100,000 children each year die of RSV infections in the developing world [5]. RSV is also an important trigger of asthma exacerbations, since half of children with RSV bronchiolitis will develop recurrent wheezing or asthma [6,7,8,9].

A genetically and antigenically closely related orthopneumovirus, bovine respiratory syncytial virus (BRSV), induces an acute lower respiratory disease in calves that shares many clinical and virological features of infant RSV-bronchiolitis [10]. Calves acquire maternally derived serum antibodies (MDA) at birth by drinking colostrum from their BRSV-immune mothers, and young calves are most susceptible to severe BRSV disease when passive immunity from MDA has decreased to moderate or low levels, usually at one month of age [11]. BRSV is estimated to cause 14 to 71% of bovine respiratory disease (BRD) outbreaks [12]. Attenuated and killed BRSV vaccines are commercially available, but their efficacy is variable particularly in calves with MDA at the time of vaccination and/or the duration of immunity is short [13,14]. There is a need to develop a safe BRSV-vaccine, which elicits a long duration of protection after one administration in the presence of BRSV-specific maternal antibodies to protect calves during a window of susceptibility that occurs when passive immunity has declined [14].

RSV vaccination in humans presents similar challenges [15]. Indeed, natural infections with RSV occur repeatedly throughout life and fail to confer complete immunity following recovery [16,17]. Ideally, RSV vaccines should give rise to better immunity than that induced by natural infection to break the virus transmission cycle, which is particularly devastating within families where infants and the elderly are both exposed to the virus. The first vaccination clinical trial in the late 1960s with FI-RSV (formalin-inactivated purified RSV in alum) failed to protect against infection and resulted in severe enhanced pulmonary disease upon seasonal RSV infection of the infant recipients (two vaccinees died and 80% were hospitalized) [18]. FI-RSV vaccine failure is explained in part by the elicitation of poorly neutralizing antibodies that can form immune complexes with RSV antigens upon natural infection leading to detrimental complement activation in airways [19]. As a consequence, the control of bronchiolitis by vaccination is a priority of the World Health Organization, and RSV vaccines have been under investigation for nearly 50 years.

The fusion (F) glycoprotein is highly conserved between RSV isolates and is the main target for neutralizing antibodies. The resolution of the structure of the F protein in either the post-fusion conformation [20] or pre-fusion conformation [21] has led to a comprehensive understanding of the neutralizing epitopes displayed on F [22]. Some neutralizing epitopes are present on both the post- and pre-fusion conformations (for instance site II and site IV) [23,24]. However, the pre-fusion form of F harbors highly potent neutralizing epitopes (sites 0 and V) not present in the post-fusion form [21]. The post-fusion form of the F protein is largely dominant on FI-RSV virions, whereas infectious RSV virions present both pre- and post-fusion F conformations [25].

Much progress has been made in the past ten years in the design of RSV subunit vaccines based on studies bringing novel insights into structural aspects of the viral proteins. For instance, the 3D-structure of some RSV proteins has been solved and stabilized immunogens produced: nanorings formed by the RSV nucleoprotein [26] as well as pre- and post-fusion states of the RSV fusion protein [20,27].

Nevertheless, full benefits of these technological advances are still hampered by inadequate knowledge of RSV pathophysiology in humans, as disease models are not available. RSV vaccine candidates are usually evaluated in the mouse, cotton rat, and nonhuman primate (NHP) pre-clinical models [28]. However, RSV is only semi-permissive in these models and thus they do not authentically represent natural infection.

Thus, BRSV infection in calves with its strong similarities to RSV infection in children appears to be highly relevant to study RSV pathogenesis and to evaluate the safety and the protective efficacy of vaccine candidates in a natural host [29]. In previous studies, we used a robust virus challenge in calves, a model which reproduces clinical signs of respiratory disease and pathological lesions similar to those observed in the field [30,31].

Using such a BRSV challenge model, the efficacy, immunogenicity, and safety of RSV vaccines can be investigated, not only for the health of cattle but also the health of infants. For instance, virus-vectored vaccine expressing human RSV proteins provided protection in seronegative calves challenged with BRSV [32].

In relation to the present study, we previously evaluated some RSV subunit vaccine candidates. The first one used nanorings formed by the nucleoprotein (N from human (H)RSV) either alone [33] or multimeric (with epitopes from the BRSV F and G proteins) in combination with recombinant BRSV phospho- (P) and matrix (M2) proteins [34]. The nucleoprotein N is highly conserved between HRSV subtypes and even bears >85% amino-acid homology with N from BRSV, so it could be an interesting component of a heterosubtypic vaccine. The second candidate used the BRSV fusion (F) protein stabilized in its pre-fusion form (preF) [31]. Both N and preF single subunit vaccines were administered twice (prime and boost) to calves purposely deprived of MDA [31,33]. However, under field conditions, calves usually carry variable to high levels of serum MDA which led us to test our candidate vaccines under these conditions. Moreover, for practical aspects of herd management (cost, labour) but also in relation to the vaccine efficacy linked to the vaccine implementation, a veterinary vaccine should be effective upon one injection.

A good vaccine candidate against human or bovine RSV should elicit a rapid and safe protective immune response in the presence of MDA (in newborn infants or young calves), preferably after a single dose. To achieve this goal, the present study was designed to combine the preF form of the BRSV F protein and HRSV N nanorings formulated in a water-in-oil adjuvant (Montanide^TM^ ISA 61 VG, Air Liquide, Paris, France) and to deliver them to calves with MDA in a single intramuscular (IM) vaccination. We demonstrated that both subunit vaccines (preF alone or preF combined with N nanorings) provided safe and highly protective immunity (at the clinical and virological levels) against an experimental challenge with BRSV. We performed an in-depth statistical analysis of all parameters recorded (virology, clinical signs of disease, immune responses) and demonstrated that neutralizing antibodies and antibodies to preF were strongly correlated with protection post challenge.

## 2. Materials and Methods

### 2.1. Calves and Vaccination Protocol

Eighteen Prim Holstein male calves born at INRAE Le Pin Research Farm (Gouffern En Auge, France) (29 December 2015–26 January 2016) were moved to the Platform of Experimental Infectiology (PFIE) INRAE Val de Loire (Nouzilly, France) at the age of 7–10 days. The quality of colostrum uptake was measured by the refractive index of the sera collected 48 h after birth. At their arrival at PFIE, all calves were bled to quantify the level of maternally derived BRSV-specific antibodies (MDA) in sera. Calves were allocated to three groups of 6 to give groups matched for age and the level of BRSV-specific serum antibodies. Each group of calves was housed in a separate room, and biosecurity measures were implemented to avoid any cross contamination. Experiments were carried out in compliance with French national rules and received the agreement number APAFIS#2719-2015110908345685v3 after approval by the Ethical Committee of Val de Loire.

At 3–7 weeks of age, calves were vaccinated intramuscularly (IM) with either 100 μg recombinant pre-fusion ‘DS2-v1′ BRSV F protein, prepared as described previously [31] and adjuvanted with Montanide^TM^ ISA 61 VG (ISA61) (kindly provided together with the IKA Ultra Turrax Tube Drive Mixer by SEPPIC S.A., Puteaux, France) in a 60/40 (v/v) proportion of adjuvant/protein in 2 mL volume per dose; 100 μg PreF with 100 μg recombinant nucleoprotein (N nanorings) in ISA61 or Phosphate Buffer Saline PBS in ISA61 (Figure 1a). N nanorings were prepared as previously described [33]. One calf (6403) allocated to the control group died suddenly three weeks after the vaccination due to an unrelated cause (suspicion of enterotoxemia).

Blood samples and nasal swabs were collected at intervals to measure antibody responses to BRSV, F, and N antigens. Blood was also collected in citrate to prepare peripheral blood mononuclear cells (PBMC) for analysis of memory T cell responses to vaccination. PBMC were cryopreserved in fetal calf serum (FCS, Eurobio, Les Ulis, France) containing 10% dimethylsulfoxide (DMSO, Sigma–Aldrich, MO, USA) and stored over liquid nitrogen until use.

One month after vaccination, all calves were challenged with 10^4^ plaque-forming units (pfu) of the Snook strain of BRSV by nebulization as described previously [30]. Clinical signs were monitored daily for one week. Nasal swabs were collected every day to establish the kinetics of virus replication. The clinical score was determined as described previously [30,34]. Seven days post-challenge, all calves were euthanized by overdose of pentobartibal (Dolethal, Vétoquinol, France). At necropsy, the lung was removed from the thoracic cage. Broncho-alveolar lavage (BAL) and lung tissue samples were collected. BAL cells, BAL supernatant, and RNA from lung tissue were prepared for further analysis as described previously [30,34].

### 2.2. Serology

BRSV-specific, N-protein-specific, and PreF-protein-specific IgG and IgA antibody titers were determined by enzyme-linked immunosorbent assay (ELISA) as described previously [31,33,35,36]. The preF ELISA assay had slight modifications: plates were coated with 200 ng of Pre-F antigen per well, and Sea Block (Thermo Scientific, Loughborough, UK) was used for saturation.

BRSV neutralizing antibodies were analysed by a modification of the method described in [37] using rBRSV-GFP (BRSV A51908 strain instead [38]) of rRSV-cherry.

For BRSV-specific MDA, BRSV-specific IgG1 antibodies were analysed using a commercial ELISA kit (SVANOVIR^®^ BRSV-Ab ELISA, Svanova, Uppsala, Sweden), in accordance with the manufacturer’s instructions, including calculations of corrected optic density (COD) and percent of kit positive control (%COD positive).

### 2.3. T-Cell Responses

BRSV-specific IFNγ-producing T cells were analysed by ELISpot using workshop cluster 1 (WC1)^+^ γδ T-cell-depleted PBMC. In brief, ELISpot PVDF membrane plates were humidified with 35% ethanol, rinsed five times with PBS, coated with mouse monoclonal antibodies specific for bovine IFNγ (MCA2112, clone CC302, Biorad, Marnes la Coquette, France) at 0.5 µg per well for 24 h at 4 °C, and blocked with 10% FCS in PBS for 2 h at 37 °C.

PBMC were thawed, and live cells were isolated by centrifugation over OptiPrep^TM (^Sigma–Aldrich, Saint Quentin Fallavier, France) with a density of 1.15 g/mL (obtained by dilution with Roswell Park Memorial Institute (RPMI) cell culture medium), at 800 g at 4 °C for 15 min, without the brake. After cell recovery and two washes of the cells by centrifugation, γδ T-cells were depleted by staining with monoclonal antibodies against the WC1 antigen (clone CC15) followed by anti-mouse IgG-coated beads (Miltenyi Biotec, Paris, France) and magnetic sorting on LD columns MACS^®^ according to the manufacturer’s instructions.

The depleted cells were thereafter washed by centrifugation, resuspended in X-VIVO™ cell culture medium (Lonza, Levallois-Perret, France) containing 2% FCS, and 100,000 cells per well were distributed in the ELISpot PVDF membrane plates. Cells were restimulated in triplicate with heat-inactivated BRSV (strain DK9402022), heat-inactivated control antigen (cell lysate from mock-infected cells), and X-VIVO™ alone or concanavalin A at 25 µg/mL, for 24 h at 37 °C. Following restimulation, the cells were lysed with water, and the plates were incubated with 0.045 µg biotin-conjugated monoclonal mouse antibodies specific for bovine IFNγ (MCA1783, Biorad, Marnes la Coquette, France) per well, overnight at 4 °C. IFNγ spots were detected by using streptavidin-alkaline phosphatase, which was incubated for 1 h at room temperature (RT), and nitro-blue tetrazolium chloride/ 5-bromo-4-chloro-3′-indolyphosphate p-toluidine salt, which was incubated for 30 min at 37 °C. Five PBS wash steps were performed after each incubation. The spots were enumerated using the AID iSPOT reader from Autoimmun Diagnostica GmbH (Eurobio, Courtaboeuf, France). The mean number of spots from the BRSV-antigen stimulated minus control antigen triplicate wells was calculated.

### 2.4. Quantification of Virus Replication

BRSV RNA coding for the F protein present in nasal secretions or in BAL cells corresponding to 10 mL of BAL was quantified by RT-PCR as previously described [36]. Median Tissue Culture Infective Dose equivalent (TCID_50_ eq.) was the unit used to express BRSV RNA levels since the standard curve in this assay was based on a BRSV-infected cell lysate with a known titer (10^5.8^ TCID_50_).

### 2.5. Histological Analysis of Lung Tissue

Microscopic analysis was performed on tissue samples from the right cranial lobes of lungs, fixed in Formalin solution (Sigma–Aldrich, Saint Quentin Fallavier, France) for 48 h, dehydrated in ethanol, and embedded in paraffin. Sections (5 µm) were deparaffinized, counterstained with hematoxylin/eosin/saffron, analysed, and photographed with a NanoZoomer (NanozoomerDigitalPathology.view software, Hamamatsu, Massy, France).

Cell subpopulation characteristics and inflammation in each section, evaluated in a blinded manner, were morphologically described. The presence and location of neutrophilic infiltrates were graded subjectively by a diagnostic pathologist considering the amount of infiltrating cells and the extent of the section affected. The scores were – (absent) no neutrophilic infiltrate detected; (+) (insignificant) few scattered solitary neutrophils detected; + (mild) small areas with infiltration of few neutrophils; ++ (moderate) multifocal areas with evident infiltration of neutrophils; or +++ (severe) widespread areas with prominent neutrophilic infiltration.

Another set of histological sections of lung tissue was stained according to Luna’s method for detection of eosinophils [39]. The sections were assessed in a blinded manner by a pathologist with no knowledge of the animal’s treatment using a Nikon Eclipse Ci-L microscope (BergmanLabora AB Stockholm, Sweden). The number of eosinophils was counted in 20 intralesional high-power fields (HPF) corresponding to 4.74 mm^2^ (40 × objective and 10 × ocular with field number (FN) 22). The number of HPF with 10–20 or >20 eosinophils was noted for each animal.

### 2.6. Label-Free Quantitative Mass Spectrometry-Based Proteomics

Proteins in BAL supernatants were analysed by tandem-mass spectrometry as previously described [40]. In brief, sample aliquots corresponding to 20 μg protein were adjusted to be dissolved in 50 mM ammonium bicarbonate, reduced with dithiothreitol (DTT), alkylated with iodoacetamide (IAA), and in-solution digested by trypsin overnight, at a trypsin:protein ratio of 1:20.

The peptides were purified by using Pierce C18 Spin Columns (Thermo Scientific, Bremen, Germany), dried, and resolved in 0.1% formic acid to a concentration of 0.3 μg/μL. They were thereafter separated by reversed phase liquid chromatography using an EASY-nLC 1000 system, a C18-column, and a 90 min linear gradient with acetonitrile. Subsequently, they were nano electrosprayed to a Q Exactive Plus Orbitrap mass spectrometer (Thermo Scientific, Bremen, Germany), and tandem mass spectrometry (MS/MS) was performed applying higher-energy collisional dissociation fragmentation on the 10 most intense peaks.

Proteins were identified and quantified by using the quantification software MaxQuant 1.5.1.2 (free software developed by Prof. Jürgen Cox at the Max Planck Institute of Biochemistry, Münich, Germany, https://www.maxquant.org) [41] and the Bos taurus proteome database extracted from Uniprot, released September 2016. The search parameters were set to Bos Taurus taxonomy, trypsin enzyme, carbamidomethylation of cysteine as static modification, and oxidation of methionine and deamidation of asparagine and glutamine as variable modifications. A decoy search database including common contaminations and reverse database was used to estimate the false discovery rate (FDR). An FDR of 1% was accepted for peptides and protein identification. The criteria for protein identification were set to at least two identified peptides per protein. The results of all samples were combined to a total label-free intensity analysis for each sample.

### 2.7. Statistical Analysis

Statistical analysis was performed with the software GraphPad Prism 8.3.1 (GraphPad Software, San Diego, CA, USA). Data shown are individual data, with mean and SD. Non parametric tests were always used: Mann–Whitney two-tailed or Wilcoxon with corrected *p*-values (Benjamini–Hochberg). As stated in the figure legend, * indicates significant differences between controls and vaccinated calves (both preF and preF+N), ^#^ indicates significant differences between the two vaccinated groups (preF versus preF+N). ^#^,* *p* < 0.05; ^##^, ** *p* < 0.01, ^###^, *** *p* < 0.001, ^####^, **** *p* < 0.0001.

Analysis of correlates of protection was done using Spearman two-tailed analysis between the immune response data versus the clinical and virological data of the 12 vaccinated calves.

For principal component analysis, R (free software, https://www.r-project.org) was used.

## 3. Results

### 3.1. Vaccination with preF or Pref+N Provides Strong Clinical Protection against BRSV Challenge

Three groups of six calves, matched for their age and level of BRSV-specific serum antibodies (Appendix A), were vaccinated at the age of three to seven weeks with preF, PreF+N or PBS (mock) formulated in ISA61 (Figure 1a). No local reaction was detected at the injection site following vaccination (Appendix A). Very mild systemic adverse effects were observed after vaccination: one1 to two calves per group showed a temperature increase of 1.5 to 2 °C the day after vaccination (Appendix A), which returned to normal two days after vaccination. One calf (6403) allocated to the control group died suddenly three weeks after the vaccination as a result of an unrelated cause.

All calves were challenged with virulent BRSV four weeks after vaccination (Figure 1a). Calves were examined daily for seven days following challenge to establish a clinical score taking into account their rectal temperature, respiratory rate, respiratory sounds, nasal discharge, cough, dyspnea, and loss of appetite, as described previously [30]. Clinical signs of disease in control calves started to increase five days after challenge (Figure 1b). Clinical signs of disease were characterized by coughing, nasal discharge, and respiratory sounds with wheezing or crackles. The calves also had slight to moderate abdominal dyspnea (four out of five control calves). Appetite was reduced in all control calves. One calf (6431) had a fever (peak temperature 40.2 °C), and another (6417) had an elevated respiratory rate (84 resp./min). In all other control calves, temperature and respiratory rate were only mildly increased (peak temperatures between 39.2 and 39.7 °C; peak respiratory rate between 40 and 64 resp./min).

There were few or no clinical signs of disease in preF- and preF+N-vaccinated calves. Accordingly, six and seven days post-challenge, controls had significantly higher clinical scores compared with calves vaccinated with preF or preF+N (Figure 1b). However, there was no significant difference between calves vaccinated with preF or pre-F+N vaccines. At post-mortem examination, all control calves had extensive macroscopic lung lesions covering 23.3 ± 7.9% of the lung surface (Figure 1c,d). In contrast, preF and preF+N-vaccinated calves had only rare lung lesions covering 0.8 ± 0.6% and 0.3 ± 0.3% of the lung surface, respectively (Figure 1c).

Thus, our data demonstrate for the first time that a single shot vaccination with preF administered in the presence of BRSV-specific MDA was sufficient to elicit potent clinical protection against a BRSV challenge. Moreover, the addition of N nanorings to the vaccine formulation did not change the level of clinical or pathological protection.

### 3.2. Vaccination Reduces the Pulmonary Inflammatory Response Following BRSV Challenge

A critical issue with RSV vaccines is the risk of priming for an immunopathological response that causes exaggerated disease upon infection. We have evaluated this issue in the present study through the monitoring of inflammation in tissues (lung sections and BAL).

As expected, seven days after challenge (the peak of clinical signs in this model, Valarcher et al., manuscript in preparation), BAL from controls contained a high proportion of granulocytes (66 ± 0.9%, mostly neutrophils) (Figure 2a). The proportion of such cells was significantly lower in BAL of preF-vaccinated and preF+N-vaccinated calves compared to controls (3.8 ± 1.1%, *p* < 0.01, and 23.0 ± 3.6%, *p* < 0.01, respectively) and was significantly lower in preF-vaccinated compared to preF+N-vaccinated calves (*p* < 0.01).

Histological examination of lung sections from cranial lung lobes confirmed the protection of vaccinated calves. Whereas four out of five controls had a marked consolidation of the lung parenchyma and a broncho- to bronchiolar-interstitial inflammatory reaction with a marked purulent exudate in the airways, such exudate could not be observed in calves vaccinated with either of the vaccines (Figure 2b). The vaccinated calves and one control calf (6434) all had a limited mild interstitial inflammatory reaction in the scarce macroscopic localized areas. These limited histologic changes were characterised by a mild infiltration of mononuclear inflammatory cells such as macrophages, lymphocytes, and plasma cells in the intra-alveolar septa.

Overall, the number of intraepithelial neutrophils and eosinophils in the airway lumen were higher in 4/5 controls compared to calves vaccinated with either of the vaccines (Table 1). One preF-vaccinated calf (6568) had a high number of eosinophils in one out of 20 examined HPF (Table 1). The histopathological inflammatory pattern in calf 6568 did not otherwise differ from the rest of the calves in this treatment group. The number of eosinophils counted in lung sections (20 examined HPF) was significantly lower in preF+N vaccinated calves compared to controls (*p* < 0.01) (Table 1).

Broncho-alveolar lavage supernatants obtained on D7 from all calves were analysed by liquid chromatography coupled to tandem mass spectrometry. The protein concentration of these samples was significantly higher in controls (average 0.60 µg/µL, range 0.49–0.74 µg/µL) compared to calves vaccinated with preF (average 0.17 µg/µL, range 0.12–0.29 µg/µL) or preF+N (average 0.22 µg/µL, range 0.10–0.52 µg/µL, *p* < 0.001). A standardised amount of protein was analysed per sample, and the number of identified proteins per calf did not differ significantly between treatment groups. The average number of identified proteins was 355 (267–433), 339 (181–483), and 316 (195–420) in controls, preF-vaccinated, and preF+N-vaccinated calves, respectively.

A total label-free intensity analysis was performed to generate relative quantities of proteins, with proteins related to neutrophils and eosinophils manually searched, and 25 neutrophil-related proteins identified (Figure 2c). Calves vaccinated with preF expressed significantly lower quantities of all these neutrophil-related proteins compared to controls (Figure 2c, *p* < 0.001). The reduction of the neutrophil-related proteins appeared less marked in preF+N-vaccinated calves (*p* < 0.05 for preF+N vs. controls), corresponding to a higher proportion of neutrophils in the BAL of preF+N compared to preF-vaccinated calves. No proteins related to eosinophils, as defined in [42] were detected, not even in calf 6568 (the calf in which a foci of eosinophils had been detected).

Thus, we can conclude that the clinical protection induced by the preF and preF+N vaccines was not associated with an adverse pulmonary inflammatory response against BRSV.

### 3.3. Vaccination with Pref Provides Strong Virological Protection against BRSV

Although the prevention of respiratory disease is a major goal of RSV vaccine candidates, breaking the transmission of the virus between individuals (whether human or bovine) is also important to decrease the disease incidence. To assess the level of virological protection in the present study, BRSV replication in nasal secretions was monitored daily by RT-PCR, and in BAL and lung tissue samples by RT-PCR.

Following BRSV challenge, BRSV-RNA was detected in nasal secretions from all five controls on D4–D7, with a peak of virus shedding in nasal secretions at D5–D6 (Figure 3a). In contrast, BRSV-RNA was only detected in nasal secretions from 2/6 preF-vaccinated calves (6568 and 6407) and from 3/6 preF+N-vaccinated calves (6561, 6562, 6570) (Figure 3a, Appendix A). The quantity of virus RNA in nasal secretions per calf from the day of challenge (D0) to the day of autopsy (D7) was estimated based on the area under the curve (AUC). Virus replication in calves vaccinated with preF or preF+N was significantly reduced compared to that in controls (Figure 3b, *p* < 0.05). BRSV-RNA was detected in BAL from all control calves, whereas no BRSV-RNA was detected in BAL from preF-vaccinated calves, and only low levels of BRSV RNA were detected in 3/6 preF+N-vaccinated calves (6561, 6562, 6570) (Appendix A). Accordingly, the quantity of BRSV-RNA was significantly lower in calves vaccinated with either of the vaccines compared to controls (Figure 3c, *p* < 0.001).

Thus, vaccination with preF and preF+N induced highly significant protection against BRSV replication both in the upper respiratory tract and the lower respiratory tract. Importantly, virus replication (estimated by area under the curve (AUC)) was nearly abrogated in preF- and preF+N-vaccinated calves.

### 3.4. Vaccination with preF and N Elicits A Strong BRSV-Specific Adaptive Immune Response Despite the Presence of Maternal Antibodies against BRSV

BRSV-specific, preF-specific, and N-specific antibodies were measured by ELISA in sera collected from all calves at the time of vaccination (week 0), challenge (week 4), and sacrifice (week 5). Antibodies were also measured in BAL fluid collected post mortem. The levels of BRSV-specific MDA first measured in sera 48 h after birth and two weeks before vaccination (D-15) had declined slightly in all groups at the time of vaccination but remained at detectable levels in all calves (Appendix A).

At the time of challenge, four weeks after vaccination, and/or on the day of post-mortem examination, five weeks after vaccination, all calves vaccinated with preF or preF+N had higher titers of BRSV-specific serum IgG1, IgG2, and IgA antibodies compared to controls (Figure 4a, *p* < 0.05–0.001). N protein-specific antibodies were elicited only in calves vaccinated with preF+N (Figure 4c). F protein-specific antibody titers were significantly higher in all calves vaccinated with preF or preF+N, compared to controls, both at the time of challenge and at post-mortem examination (Figure 4b). In summary, except for N-specific antibodies, there were no significant differences in IgG antibody responses between the two vaccinated groups (Figure 4a–c).

Neutralizing antibodies were present at moderate levels (median = log_10_ 2.4 ± 0.7) in the serum of all calves at the time of vaccination (Figure 4d) and had declined to the limit of detection in the control group at the time of challenge. In contrast, both preF- and preF+N-vaccinated calves had significantly higher neutralizing antibody titers at the time of challenge (*p* < 0.001) and post-mortem examination (*p* < 0.001) compared to controls (Figure 4d).

The mucosal IgA response induced by vaccination was measured in nasal secretions at weeks four and five and in BAL at week five. In contrast to the control group, both vaccinated groups (preF and preF+N) had BRSV-specific IgA in nasal secretions one week after challenge (Figure 5a, *p* < 0.001 versus controls). High titers of BRSV-specific IgA were detected in BAL from all six preF+N-vaccinated calves one week after challenge (Figure 5b). In contrast, BRSV-specific IgA was detected in low titers in the BAL of only three calves vaccinated with preF alone and in none of the controls (Figure 5b). The high titers of BRSV-specific IgA elicited in preF+N-vaccinated calves can be related to the presence of both F- and N-specific IgA (Figure 5b).

BRSV-specific T-cell responses were analysed by IFNγ-ELISpot after stimulation of WC1^+^ γ/δ^+^-depleted PBMC with heat-inactivated BRSV or control antigen two weeks after vaccination (pre-challenge) and six days after BRSV challenge. Pre-challenge, BRSV-specific IFNγ-producing cells (ipc) were detected in five out of six preF-vaccinated calves and in all six preF+N-vaccinated calves (Figure 4e). After challenge, all vaccinated calves had more than 10 ipc/10^5^ cells, with values ranging from 34 to 181 ipc in the preF group and from 14 to 176 ipc in the preF+N group. In contrast, only a few BRSV-specific IFNγ-producing cells (< 10 ipc/10^5^ cells) were detected in controls prior to and after challenge. Altogether, the numbers of IFNγ-producing cells following BRSV challenge were similar to those seen prior to challenge (Figure 4e).

In conclusion, both vaccines elicited a broad range of BRSV-specific immune effectors at systemic and mucosal sites.

### 3.5. Correlation of Protection against BRSV Infection with Immune Effectors Induced by Vaccination

To gain further insight into the immune parameter(s) associated with protection against BRSV induced by the subunit vaccines, we took advantage of the fact that the two subunit vaccines induced a relatively wide range of values for immunological, virological, and clinical parameters (21 parameters measured for 17 calves). These parameter values were loaded in a principal component analysis (PCA) in order to identify the main axes of the data variance as well as the key variables. In a first analysis where data from the control and vaccinated groups were used, the Dim 1 axis, which explained 54.65% of the variance, clearly separated the data from the control group and the two vaccinated groups, with the data from the two vaccinated groups overlapping on the x-axis (Figure 6a).

In a second analysis using the same group data (357 individual values), the PCA loading from each parameter showed that the clinical, inflammation, and virological scores (red arrows) clustered together tightly (except for the eosinophilic scoring in lung tissue) and opposed the immunological parameters pre- or post-challenge (black arrows for immune responses pre-challenge and blue arrows for immune responses post-challenge) (Figure 6b). The strongest opposition was with neutralizing antibodies either pre- or post-challenge, then preF IgG pre- or post-challenge, and finally BRSV-IgG2 post-challenge. IgA to BRSV whether measured in BAL or in sera did not oppose the disease-related parameters. The IFNγ-T-cell responses pre- and post-challenge were materialized by shorter arrows, thus indicating less weight in the group separation (Figure 6b).

In a third analysis, only parameters collected from the vaccinated calves were considered (252 individual values). The PCA showed that the two vaccinated groups can be separated according to Dim 1 axis, but this axis only explained 24.45% of the variance (Figure 6c). The parameters that contributed most to the separation between the two groups are shown by the arrows in the lower right quadrant with longest projections on the x-axis (Figure 6d). They were BRSV-IgA in serum (pre and post-challenge) and BAL, as well as % polymorphonuclear leukocytes (PMN) in BAL (Figure 6d).

To identify markers potentially predictive of protection, we performed a one-to-one correlation analysis (Spearman test, two-tailed) between the clinical/virological parameters and the immunological parameters (measured at four weeks after vaccination, before challenge) for all vaccinated calves (preF and preF+N) but not the mock-vaccinated calves. Only the significant correlations (*p* < 0.05) and the correlations with *p* < 0.1 are shown (Figure 6e). Virus replication in nasal secretions (RNA nasal AUC) and in BAL (RNA BAL) were negatively correlated with PreF IgG titers and nAb titers, respectively (Figure 6e). Interestingly, the presence of eosinophils in lungs was negatively correlated with BRSV IgA titers in serum whereas a positive correlation was observed for % of PMN in BAL.

Together, the global analysis of the immune, virological, and clinical data of calves vaccinated with preF or preF+N indicated that BRSV-neutralizing Ab and preF Ab are associated with virological protection and suggest that these Abs were instrumental in the protective immunity induced by our subunit vaccines.

## 4. Discussion

RSV-induced respiratory diseases in humans and cattle can be envisaged as a One Health challenge, since the obstacles to overcome to obtain safe and effective vaccines are common to calves and infants. In fact, vaccines against RSV are urgently needed to reduce the burden of bronchiolitis in infants, and none of the licensed BRSV vaccines are fully effective, particularly in young calves with BRSV-specific MDA at the time of vaccination [14]. Moreover, bovine vaccines often necessitate several administrations to reach and maintain a sufficient level of clinical protection.

In the present study, we provide evidence that single-shot pre-F and N subunit-based vaccines can protect calves vaccinated in the presence of BRSV-specific MDA. This can be considered a major breakthrough, as it is the first report of complete protection from BRSV challenge under vaccination conditions close to those in the field.

The first vaccine trial was previously performed with the same subunit bovine preF (DS2), administered with a prime and a boost in calves deprived of colostrum at birth (thus without MDA) [31]. The BRSV F protein was stabilized in its pre-fusion state using the same strategy of mutations as for the human RSV pre-fusion protein [27]. In this previous vaccine trial, pre- and post-fusion states of F were compared for their ability to generate protective immunity in calves. The preF immunogen proved much better than postF at inducing BRSV-neutralizing antibodies, along with clinical and virological protection [31]. Likewise, in a recent phase I clinical trial involving vaccination of healthy human adults with DS-Cav1 (stabilized human preF), a large rise in RSV neutralizing titers was reported [43].

In the present vaccine trial, a single IM vaccination with either preF or preF+N adjuvanted with ISA61 (water in oil emulsion) induced a strong immune response as well as significant clinical and virological protection against BRSV challenge.

The levels of preF-IgG and nAbs elicited in sera by vaccination before challenge (four weeks post vaccination) were the parameters that most negatively correlated with virological parameters, meaning that they were associated with virological protection. In contrast, preF IgA titers in BAL or in sera did not show significant correlations with the virological parameters. These observations are consistent with recent epidemiological studies in Nepalese infants and their mothers, which showed a correlation between breast milk preF IgG antibodies (but not preF IgA) and the occurrence of acute RSV illness in infants [44]. Another study documented in a small cohort of infants analysed the relationships between serum PreF-, PostF-, and G-specific antibodies and the degree of severity of acute RSV illness. PreF antibodies were detected at higher concentrations than postF and G antibodies in infants less than four months of age, with concentrations of these antibodies declining rapidly after four months, indicative of their maternal origin [45]. High concentrations of preF or G IgG antibodies were associated with less severe RSV illness, whereas postF antibody titers were not different between mild or severe RSV illness [45]. Moreover, antibodies to preF are the most abundant and the most effective at neutralizing RSV [45,46]. Thus, a mounting body of evidence in both infants and calves suggests that preF IgG is a strong correlate of protection against RSV and is associated with potent RSV-neutralizing capacity.

Importantly, in the present study, calves were born from mothers vaccinated IM with a live-attenuated BRSV vaccine (Rispoval RS-BVD, Zoetis, France) at the end of their gestation (at eight months) as part of routine husbandry procedures. At seven months of gestation (before Rispoval vaccination), all cattle had BRSV antibodies (mean ± SD of 66 ± 21% COD of the positive BRSV serum provided with the Svanova commercial kit). Thus, MDA in calves was probably enriched in BRSV antibodies elicited upon vaccination of cattle. Whether these antibodies reacted mostly to preF or postF is not known. It was recently shown that the infant antibody repertoire differs substantially from the human adult repertoire, including in the patterns of epitope recognition [47]. If MDA was directed toward antigenic sites not frequent in calf repertoire, it may explain why preF vaccination succeeded in priming a strong antibody response in calves. In addition, the adjuvant chosen for the study, ISA61, is known to elicit strong and long lasting cellular and antibody responses in cattle [48,49], which may have contributed to overcoming the MDA inhibitory effect on the vaccine immune response under our experimental conditions.

As shown in Appendix A, MDA titers declined rapidly after birth but were still detectable at the time of vaccination, and so were nAbs (Figure 4d, week 0, Log_10_ nAb titers 2.50 ± 0.7, mean ± SD). One calf, 6434, in the control group had neither lung lesions (Figure 1c) nor detectable virus replication in nasal secretions (Figure 3b) but had sustained titers of BRSV-specific IgG antibodies from the day of vaccination up to two weeks after vaccination (Appendix A). Thus, one can hypothesize that this particular calf in the control group was protected by high MDA levels.

There was little if any difference between the two subunit vaccine formulations in terms of efficacy and safety as shown by the PCA (Figure 6). Our initial hypothesis was that N nanorings would provide BRSV-specific cellular immunity as previously shown [33,50]. However, preF by itself appeared sufficient to prime BRSV-specific IFNγ cellular responses (Figure 4e). The capacity of the fusion protein of RSV to prime specific T cell polyfunctional responses has been recently characterized following prime-boost vaccination of macaques with adenoviral vectors encoding F [51]. To our knowledge, the present study is the first demonstration that the preF subunit vaccine by itself is potent at inducing T cell responses. Adding N nanorings to the vaccine formulation did not increase the numbers of BRSV-specific IFNγ secreting cells in PBMC before or after challenge (weeks four and five, Figure 4e).

The presence of N nanorings together with preF in the vaccine formulation resulted in stronger mucosal BRSV and preF IgA responses (BAL and nasal secretions) measured post-mortem (Figure 5) and a greater inflammatory/neutrophilic reaction in BAL than that seen in preF-vaccinated calves (Figure 2a,c). Consequently, IgA responses measured before and after challenge (weeks 4,5), and post-mortem %PMN in BAL (week 5) were the most important parameters contributing to the separation between the two vaccinated groups (Figure 6c,d). In humans, neutrophils express the IgA receptor FcαR (CD89), and its cross-linking by immune complexes can lead to their activation and ability to eliminate infected cells but can also lead to enhanced tissue inflammation, a mechanism that could explain the apparent association between IgA and neutrophil levels in BAL in our study [52]. Whether such pathways apply to bovine neutrophils remain to be explored, since a bovine FcαR closely related to CD89 has been identified [53]. Conversely, it has also been proposed from in vitro studies that activated neutrophils via secretion of TGF- increase the expression of pIgR on epithelial cells, which in turn increases translocation of IgA into secretory IgA in the airway lumen [54]. This implies that the positive association we observed under our experimental conditions between IgA and PMN may not be necessarily deleterious and could even contribute to virus control. It is striking too that neutrophils did not appear to accumulate in lung sections of individuals vaccinated with preF + N (Table 1). One can hypothesize that once activated, neutrophils migrate rapidly from the blood vessels to the alveolar space as proposed during the course of bronchiolitis [55].

Thus, N nanorings did not improve vaccine efficacy in terms of protection and immune responses and conversely were associated with a mild inflammation in airways at post-mortem. This makes them dispensable from the vaccine formulation. Alternatively, N is a good antigen for a BRSV-DIVA test using preF as a vaccine antigen. We monitored antibody reactivity to N and whole BRSV antigens in sera of naturally/experimentally infected cattle/calves with or without MDA. We could demonstrate a good fit between N reactivity and reactivity to whole BRSV antigens (unpublished data).

The safety of preF vaccination was shown post-mortem by the absence of an inflammatory reaction in the lungs, as demonstrated by the low percentage of granulocytes in BAL cells, the reduced cellular infiltration in lungs, and nearly undetectable inflammatory proteins in BAL fluid in comparison with mock-vaccinated calves (Figure 2). Only in one calf (6568) vaccinated with preF was a foci of eosinophils detected in lung sections post-mortem. A pathway analysis was performed on 27 proteins that had been identified in highest quantities in this calf compared to all other calves (Ingenuity Pathway Analysis, content version 43605602, release date 03 28 2018). The identified top canonical pathway was acute phase response signaling (*p* = 3.20 E-12, overlap 4.7%, eight proteins identified out of 170 proteins in pathway). Among the 27 proteins identified in highest quantities in this calf, plasminogen, kininogen-1, plasma kallikrein, and albumin were identified that might be related to bradykinin production, resulting in airway microvascular leakage and eosinophil or neutrophil recruitment. This calf did not show abnormal clinical scores nor delayed virus clearance, suggesting that an exacerbated reaction had not occurred in this animal, but rather that the response was related to virus clearance or an unrelated event.

It will be necessary to extend these findings further by investigating the duration of immunity compared to commercial vaccines and other safety aspects. Indeed lower doses of BRSV preF could prime for exacerbated pulmonary pathology as has been shown in small animal models of human RSV [56]. Field trials will be important to explore other vaccine issues. The pre-F vaccine should provide cross protection against different field strains of BRSV as the BRSV F protein is highly conserved between different isolates. Nevertheless, the level of cross-protection against other field strains of BRSV should be confirmed. In addition, a possible gender effect on vaccine response should be explored. Compared to commercially available vaccines, this vaccine offers an advantage since it can be administrated as a single dose by the parenteral route. Indeed, failure of protection post vaccination for vaccines already on the market can be observed when the vaccination is not performed adequately: correct administration of the vaccine by the intranasal route or forgetting to revaccinate when a boost is required to obtain a sufficient level of protection.

## 5. Conclusions

Based on these data, we can conclude that the subunit preF based vaccine is a very promising vaccine. Indeed, with only one shot and despite BRSV-specific maternal immunity, it induced rapid and safe protection against experimental BRSV challenge. If this vaccine induces long lasting protection, it will probably constitute a good option to reduce the burden of BRSV disease in calves and may even be considered as a promising vaccine strategy for infants.

## Figures and Tables

**Figure 1 vaccines-08-00231-f001:**
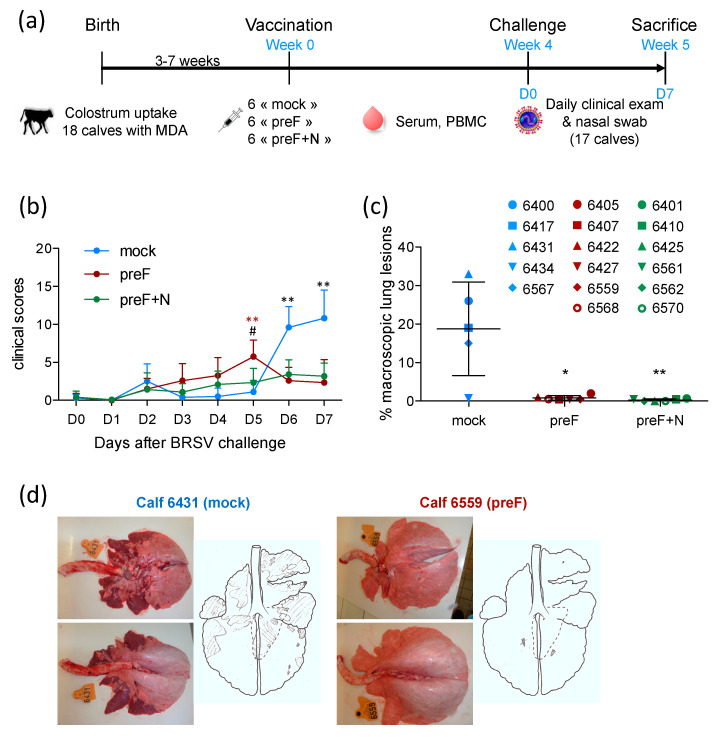
Vaccination with preF or preF+N provides strong clinical protection against BRSV challenge. (**a**) Study design. D0 is the day of challenge (4 weeks after a single vaccination). Calves were sacrificed 7 days after challenge (D7), 5 weeks after vaccination. (**b**) Monitoring of clinical protection against BRSV infection of vaccinated calves (3 groups of 5–6 animals). The clinical scores were calculated as described in [30] based on daily clinical examinations of calves. (**c**) Macroscopic lung lesions are expressed as a % of total lung surface calculated from photographs of the lungs made post mortem (**d**) (ImageJ v1.52a free software, https://imagej.nih.gov/ij/index.html). Each spot represents one calf (controls/blue; preF/red; preF+N/green). Mean values with standard error of the mean are indicated by the black line. (d) Ventral and dorsal aspects of the lung were photographed. Ventral and dorsal lesions were also drawn onto a lung scheme. Statistics are Mann–Whitney two-tailed (b,c), day by day comparison of groups (b), * indicates significant difference between controls and vaccinated calves (black stars: mock versus both preF and preF+N, red stars: mock versus preF), ^#^ indicates a significant difference between the two vaccinated groups (preF versus preF+N). (* *p* < 0.05, ** *p* < 0.01).

**Figure 2 vaccines-08-00231-f002:**
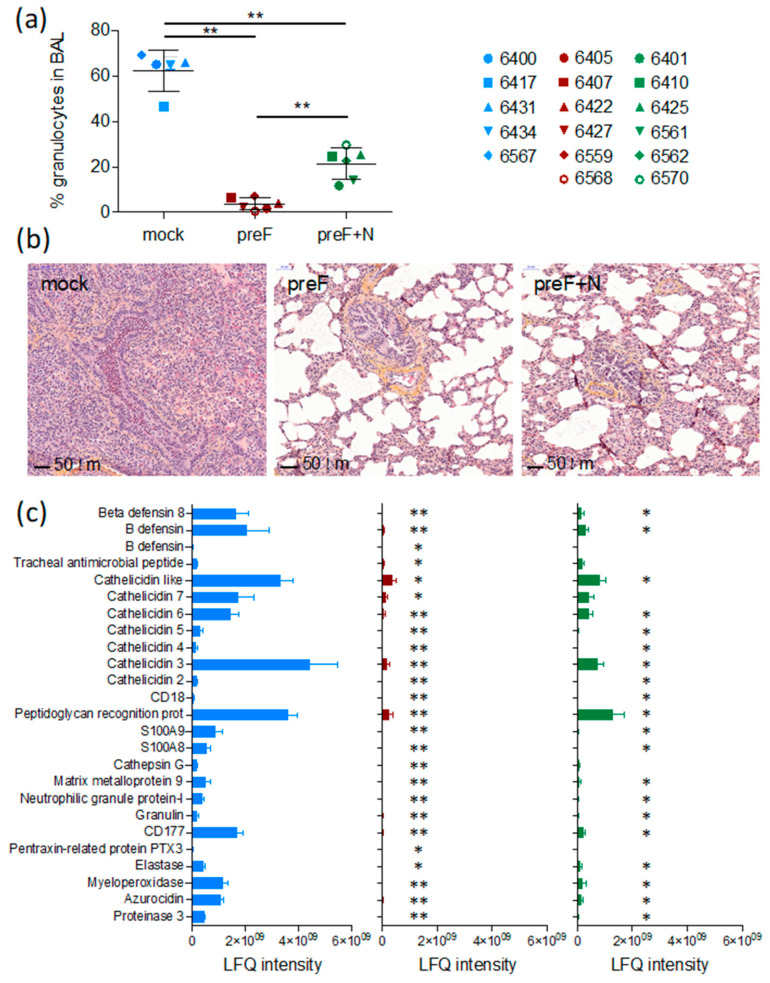
Safety and efficacy of preF vaccines according to their inflammation profiles in BAL and lungs. (**a**) % of granulocytes counted in BAL upon May–Grünwald–Giemsa staining. Each spot represents one calf (controls/blue; preF/red; preF+N/green). Mean values with standard error of the mean are indicated by black line. Statistics are Mann–Whitney two-tailed values (* *p* < 0.05, ** *p* < 0.01). (**b**) Biopsies were taken from the cranial lobe, and lung sections were stained with H&E. (**c**) Broncho-alveolar lavage (BAL) supernatants from vaccinated calves and controls, obtained on D7 after BRSV challenge, were analysed by liquid chromatography coupled to tandem mass spectrometry (the shotgun method, LC-MS/MS, Orbitrap, Thermo Scientific, Bremen, Germany). Proteins were identified, and a total label-free intensity analysis was performed using MaxQuant 1.5.1.2 software. Twenty-five proteins were selected based on being related to neutrophils, and the relative quantities of these are presented as label-free quantification (LFQ) intensities. Statistical tests are Wilcoxon, with corrected *p*-values (Benjamini–Hochberg). Comparisons between groups were done for each of the 25 proteins (mock versus preF or mock versus preF+N). (* *p*<0.05, ** *p*<0.01).

**Figure 3 vaccines-08-00231-f003:**
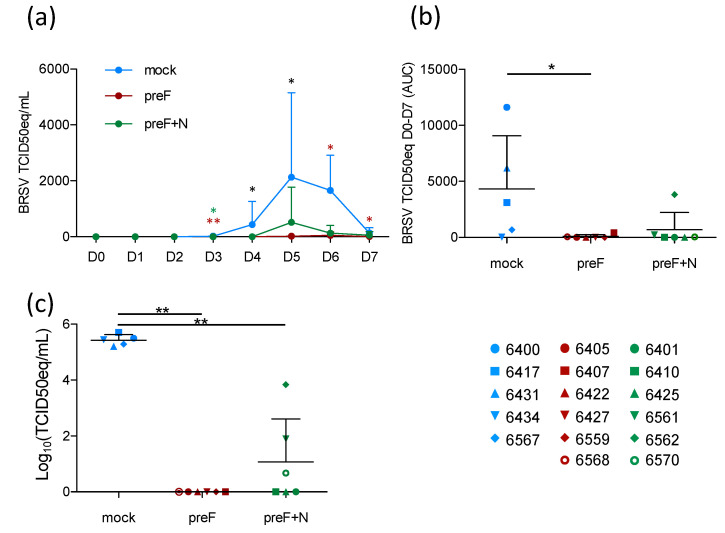
Vaccination with preF provides strong virological protection against BRSV in upper and lower respiratory tracts. Nasal secretions were collected daily after challenge, from D0 to D7 (before autopsy). BAL cells were collected post-mortem. Total RNA was extracted, and BRSV-RNA detected by real time PCR, expressed as TCID50eq. The cut off of the RT-PCR assay is 0.7 TCID 50eq. (**a**) Kinetics of virus replication in the nasal passages. Mean values with standard deviation are plotted (controls/blue; preF/red; preF+N/green). (**b**) Area under the curve (AUC) for total virus RNA D0-D7. (**c**) BRSV-RNA in BAL cells collected post mortem. (b–c) Each spot represents one calf (controls/blue; preF/red; preF+N/green). Mean values with standard error of the mean are indicated by black lines. Statistics are Mann–Whitney two-tailed, day by day comparison of groups (a), * indicates a significant difference between controls and vaccinated calves (black stars: mock versus both preF and preF+N, red stars: mock versus preF, green stars: mock versus preF+N). (* *p* < 0.05, ** *p* < 0.01, *** *p* < 0.001).

**Figure 4 vaccines-08-00231-f004:**
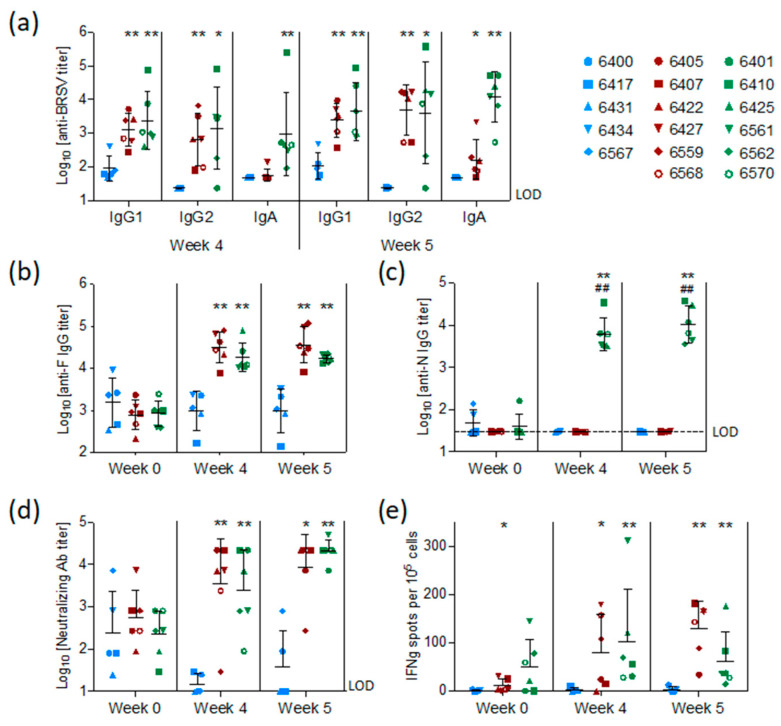
Systemic antibody and T cell responses elicited by pre-F or pre-F and N. Calves were vaccinated IM with preF or preF+N in ISA61 adjuvant or mock (ISA61 alone) and were challenged 4 weeks later with BRSV. (**a**) BRSV-specific IgG1, IgG2, IgA titers in sera at weeks 4 and 5 after vaccination. (**b**) pre-F-specific Ab titers in sera at weeks 0, 4, and 5 after vaccination. (**c**) N-specific Ab titers in sera at weeks 0, 4, and 5 after vaccination. (**d**) BRSV-neutralizing Ab titers in sera at weeks 0, 4, and 5 after vaccination. (**e**) IFNγ BRSV-specific T cells among PBMC depleted from γδ^+^ T-cells at weeks 4 and 5 after vaccination counted by ELISPOT. Each spot represents one calf (controls/blue; preF/red; preF+N/green). Mean values with standard error of the mean are indicated by the black line. Statistical analysis was performed by the non-parametric test Mann–Whitney, two-tailed *p* value. * indicates a significant difference between controls and the preF- or preF+N-vaccinated calves, # indicates a significant difference between the two vaccinated groups (preF and preF+N). (* # *p* < 0.05, ** ## *p* < 0.01). Limits of detection are log10 0.5 for N-specific IgG, log10 1.4 for BRSV-specific IgG2, log10 1.7 for BRSV-specific IgA and IgG1, and log10 1.0 for BRSV neutralising antibodies.

**Figure 5 vaccines-08-00231-f005:**
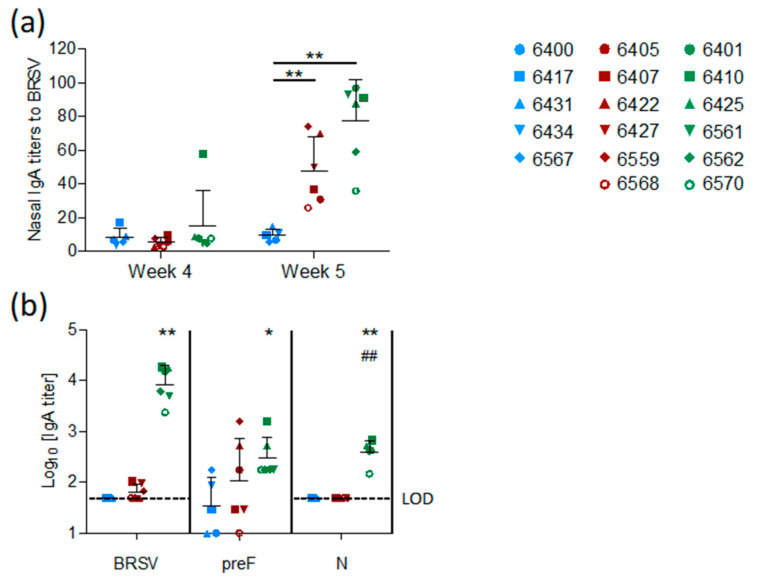
Mucosal antibody responses elicited by pre-F or pre-F and N. Calves were vaccinated IM with preF or preF+N in ISA61 adjuvant or mock (ISA61 alone) and were challenged 4 weeks later with BRSV. (**a**) BRSV-specific IgA in nasal secretions sampled on the day of challenge (week 4) and one week after challenge (week 5). (**b**) BRSV-, preF-, and N-specific IgA in BAL supernatant was collected post-mortem (week 5). Each spot represents one calf (controls/blue; preF/red; preF+N/green). Mean values with standard error of the mean are indicated by the black line. Statistical analysis was performed by the non-parametric test Mann–Whitney, two-tailed *p* value. * indicates a significant difference between controls and the preF- or preF+N-vaccinated calves, # indicates a significant difference between the two vaccinated groups (preF and preF+N). (* # *p* < 0.05, ** ## *p* < 0.01).

**Figure 6 vaccines-08-00231-f006:**
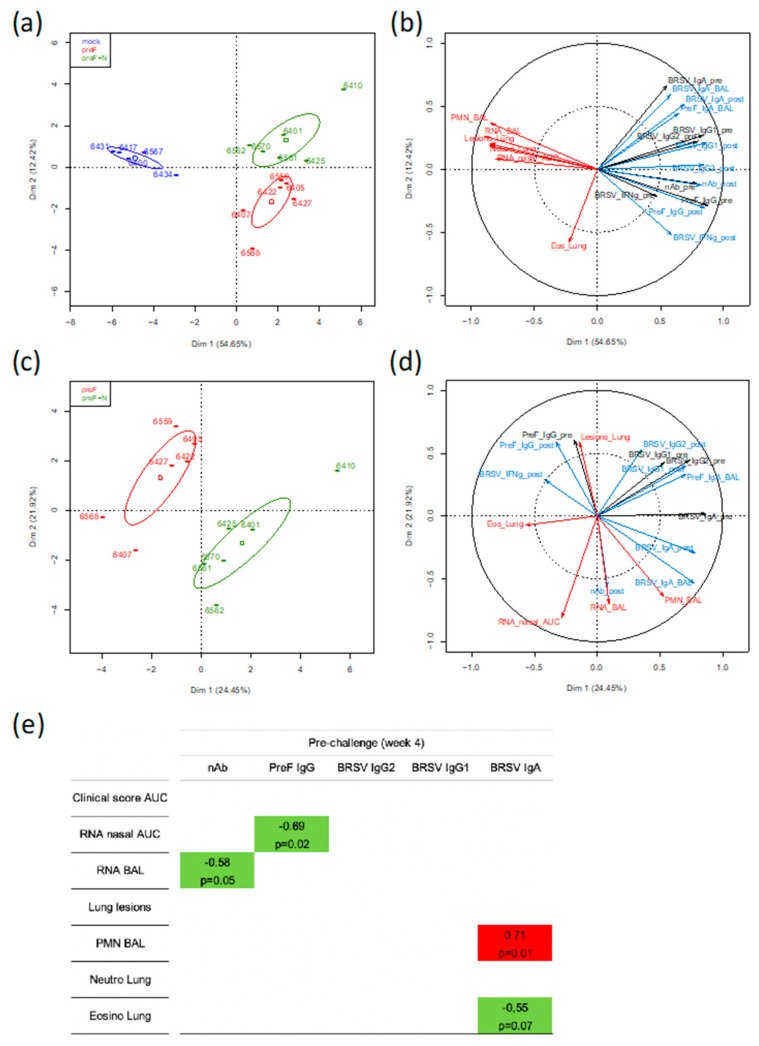
Principal component analysis (PCA) and correlation analysis of the clinical, virological, and immunological response data of the calves vaccinated with preF, preF+N or mock-vaccinated. Calves were vaccinated IM with preF or preF+N in ISA61 adjuvant or mock (ISA61 alone) and were challenged 4 weeks later with BRSV and sacrificed one week after challenge. (**a**) The PCA plot of the clinical, virological, and immunological response data (21 PCA variables) of vaccinated and non-vaccinated calves is depicted, with each calf represented as a dot plus ID number in a specific color according to its group assignment: Dim1 explained 54.65% of the total variation of the data between calves, and Dim 2 explained a further 12.42% of the variation. The vaccination regimen is indicated by a distinct color (blue for controls, red for preF, and green for preF+N). (**b**) 357 individual input values were loaded (21 parameters measured on 17 calves, see Materials and methods), and the most relevant ones are indicated on the figure for clarity (R > 0.5). In red the virological, inflammatory & clinical scores; in black the immunological parameters measured at week 4 (pre-challenge), and in blue the immunological parameters measured at week 5 (post-mortem). (**c**) Same as (a) but considering only the 252 individual data collected from the 12 vaccinated calves (preF and preF+N). (**d**) Same as (b) but considering only the 252 individual data collected from the 12 vaccinated calves (preF and preF+N). (**e**) Spearman Correlation analyses, two-tailed, between the immune response data collected pre-challenge (week 4) versus the clinical and virological data of the 12 vaccinated calves. Only significant/nearly significant correlations are shown, in green where negative and in red where positive.

**Table 1 vaccines-08-00231-t001:** Presence of polymorphonucleated cells in lung sections

Group	ID	Neutrophil Score ^1^			Eosinophil
		Intraepithelial	Bronchi/Bronchioli Lumen	Alveoli Lumen	Number ^2^
ISA61	6400	+++	+++	++	21
	6417	+++	+++	++	32
	6431	+++	+++	++	10
	6434	−	−	−	73
	6567	+++	+++	++	26
preF	6405	−	−	−	4
	6407	(+)	−	−	13
	6422	(+)	−	−	4
	6427	−	−	−	3
	6559	−	+	+	6
	6568	(+)	(+)	−	152
preF+N	6401	−	−	−	3
	6410	(+)	−	−	2
	6425	−	−	−	1
	6561	−	(+)	+	16
	6562	−	−	−	3
	6570	−	−	−	7

^1^ Grading of neutrophilic infiltrate in lung tissue. − absence, (+) insignificant, + mild, ++ moderate, +++ severe. ^2^ Sum of eosinophils in 20 high-power fields (area of 4.74 mm^2^). The difference between groups ISA61 and preF+N is significant (** *p* < 0.01), Mann–Whitney tests, 2-tailed.

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
