# Peer review of "A Single Shot Pre-fusion-Stabilized Bovine RSV F Vaccine is Safe and Effective in Newborn Calves with Maternally Derived Antibodies"

_vaccines, 2020, doi:10.3390/vaccines8020231_

Round 1

Reviewer 1 Report

This manuscript talks about the evaluation of a single shot preF subunit vaccine candidate in calves with maternally derived serum antibodies (MDA) against BRSV challenge and highly protective immunity have been observed. The results have been presented clearly, and the conclusions are supported well by the data. Here are some aspects that this review concerns:

Major comments:

  1. In the introduction part, the authors talk about BRSV too late. It makes this reviewer thinking that this manuscript is to test a hRSV vaccine candidate using calves as the animal model. So it is better to introduce earlier and more about BRSV.

         Can calves be the good animal models for testing hRSV vaccine?

  1. Are all the calves used in this study BRSV negative? If yes, why the authors hypothesized that calf ID6434 may have BRSV infection before being moved to the facility (Line 490)? Did the authors take nasal swabs of all calves to test BRSV-RNA?
  2. In figure 1C, please provide the representative pictures of microscopic lung lesion after challenge for each group.
  3. Do authors think about using a different challenge strain that is far from the immunization strain in the phylogenetic tree, like BRSV strain A51908 or 375?

Minor comments:

  1. Line 138, please provide more information about prefusion “DS2 BRSV F protein”. Is it DS2-v1?
  2. Line 164, is the virus stain used to make rBRSV-GFP same as the challenge strain? Please provide the train information in the manuscript.
  3. Figure 4e week 0, move the “*” to the top of preF +N group. 

Author Response

Major comments:

1- In the introduction part, the authors talk about BRSV too late. It makes this reviewer thinking that this manuscript is to test a hRSV vaccine candidate using calves as the animal model. So it is better to introduce earlier and more about BRSV. Can calves be the good animal models for testing hRSV vaccine?

The introduction was modified accordingly. In the revised introduction, bovine RSV arrives line 56 just after human RSV. Vaccine issues are discussed together because they present same challenges: 1) bovine vaccine then 2) human vaccine.

Finally we put more emphasis about the use of calves to test human RSV vaccines (line 104-107). And quoted a new ref 32 (Taylor et al. Efficacy of a virus-vectored vaccine against human and bovine respiratory syncytial virus infections. Science Translational Medicine 2015:Vol. 7).

2- Are all the calves used in this study BRSV negative? If yes, why the authors hypothesized that calf ID6434 may have BRSV infection before being moved to the facility (Line 490)? Did the authors take nasal swabs of all calves to test BRSV-RNA?

We did not take nasal swabs of the calves before moving them to the experimental bioconfined facilities. In fact, all calves are born in the same farm with intensive BRSV vaccination practices. Thus all cattle are seropositive. We cannot exclude that BRSV circulated in the herd but then several calves would have been affected which was not the case. Since we have no nasal samples to test our hypothesis and because there was no anamnestic response after challenge, we have decided to remove this misleading hypothesis from the discussion (line 493).

3- In figure 1C, please provide the representative pictures of microscopic lung lesion after challenge for each group.

Representative pictures of microscopic lung lesions are shown Figure 2b. Thus we assumed that the reviewer was referring to representative pictures of macroscopic lung lesions depicted as % area on graph Figure 1c. Thus we have added Figure 1d, representative photographs and drawings of macroscopic lung lesions, one calf from the mock group and one calf from the preF group.

4- Do authors think about using a different challenge strain that is far from the immunization strain in the phylogenetic tree, like BRSV strain A51908 or 375?

BRSV strains and their comparisons are made via the G gene, which varies among strains, in contrast to other genes such as the F gene (Valarcher et al. 2000). When comparing the aa sequences of the PreF (Strain 391.1), to the sequences of the challenge virus (snook) and to 375 and A51908 suggested by the reviewer 1, we find few substitutions in the main antigenic sites ø (amino acid residues 195-210 and 61-68: one mutation at residue 202 E->K for strain 375) and antigenic site II (amino acid residues 254-280: one conservative mutation at residue 260 L->I for strain A51908), suggesting that our candidate vaccine will induce an equal level of protection against the latest strains. Of note, the sequence of the F protein of the strain 375 is incomplete, but the missing residues correspond to the C-terminal region of F which does not contain antigenic sites but correspond to the trans-membrane domain and cytoplasmic tail.

Thus we expect that the preF vaccine will provide cross-protection against circulating strains, even if further evaluation in the field will be needed. A statement was added in the discussion line 546.

Minor comments:

1- Line 138, please provide more information about prefusion “DS2 BRSV F protein”. Is it DS2-v1?

Yes it is. It has been corrected line 143 in M&M.

2- Line 164, is the virus stain used to make rBRSV-GFP same as the challenge strain? Please provide the strain information in the manuscript.

rBRSV-GFP was made using the A51908 strain of BRSV (Goris et al., 2009, ref 38). The text has been modified accordingly line 169.

3- Figure 4e week 0, move the “*” to the top of preF +N group. 

The “*” is at the correct place. The difference between mock and preF +N is indeed not significant whereas it is significant between mock and preF (Mann Whitney, two-tailed).

Reviewer 2 Report

In this manuscript, Riffault et al, have extensively studied a single-shot vaccination strategy using two subunit vaccines, the bovine (B)RSV pre-fusion F (preF) alone or with nanorings formed by the RSV nucleoprotein (preF+N) adjuvanted with ISA61VG to reduce the burden of BRSV disease in calves with MDA. Overall, this study contributes good knowledge to existing literature on BRSV vaccines and open new directions for single-shot vaccine development. Authors need to address some minor concerns before considering this work for a publication:

  1. Why did authors choose to include only ISA61VG (mock) as control rather than a sterile diluent control (like PBS)?
  2. Did authors perform any detailed examination of the lung lesion by lobe?
  3. Authors have only tested in 6 calves (n=6). It is worth commenting somewhere in text how these results can be extrapolated for future development for this vaccine strategy.
  4. In the current manuscript, vaccine efficacy was studied on Male calves. Do authors expect any sex-based differences in vaccine efficacy?

Author Response

1- Why did authors choose to include only ISA61VG (mock) as control rather than a sterile diluent control (like PBS)?

For our mock control group, we used PBS with ISA61VG, prepared like the subunit vaccines in which PBS was the diluent (as explained in M&M lines 144-147). We believe it is important to include the adjuvant in the control group in case it creates some kind of innate immune background before challenge. Although it would have been interesting to test this issue, we could not accommodate a fourth group of 6 calves treated with PBS only in our bioconfined facilities.

2- Did authors perform any detailed examination of the lung lesion by lobe?

Yes we did. We have now provided a photograph and drawing of the lung lesions (Figure 1d).

3- Authors have only tested in 6 calves (n=6). It is worth commenting somewhere in text how these results can be extrapolated for future development for this vaccine strategy.

It will be necessary to expend further these findings by investigating the duration of immunity (paper in preparation) compared to commercial vaccine and other safety aspects for instance using reduced dose of antigens (paper in preparation) before going further to perform field trials.

Compared to what is on the market, this vaccine will probably enable a better vaccine efficacy by making easier the vaccination process since it has to be administrated only once by parenteral route in calves with or without BRSV MAD. Indeed, failure of protection post vaccination for vaccines already on the market can be observed when not performing the vaccination adequately: administrate properly the vaccine by intranasal route or forgetting to revaccinate when a boost is required to get a sufficient level of protection.

This has been added to the discussion lines 543-552.

4- In the current manuscript, vaccine efficacy was studied on Male calves. Do authors expect any sex-based differences in vaccine efficacy?

This is an important issue. In fact in the present study as for other previous studies, males only were available on the market for purchase for our experiment since female are kept for production.

To the best of our knowledge, differences according to sex have not been reported in vaccine response to BRSV or in other bovine vaccines. Interestingly, such gender effect on the vaccine response has been reported in pigs vaccinated against PRRSV (personal communication).

Thus we have added at the end of the discussion line 547 : “It will be necessary to expend further these findings by investigating the duration of immunity compared to commercial vaccines and other safety aspects before going further to perform field trials in which other aspects will be explored such as cross protection against field BRSV strains or the gender effect on vaccine response”.

This will answer point 4 of reviewer 1, as well as points 3 and 4 of reviewer 2.
